# Large differences in regional precipitation change between a first and second 2 K of global warming

Peter Good[1], Ben B.B. Booth[1], Robin Chadwick[1], Ed Hawkins[2], Alexandra Jonko[3] & Jason A. Lowe[1]

For adaptation and mitigation planning, stakeholders need reliable information about regional precipitation changes under different emissions scenarios and for different time periods. A significant amount of current planning effort assumes that each K of global warming produces roughly the same regional climate change. Here using 25 climate models, we compare precipitation responses with three 2 K intervals of global ensemble mean warming: a fast and a slower route to a first 2 K above pre-industrial levels, and the end-of-century difference between high-emission and mitigation scenarios. We show that, although the two routes to a first 2 K give very similar precipitation changes, a second 2 K produces quite a different response. In particular, the balance of physical mechanisms responsible for climate model uncertainty is different for a first and a second 2 K of warming. The results are consistent with a significant influence from nonlinear physical mechanisms, but aerosol and land-use effects may be important regionally.

[1] Met Office Hadley Centre, Exeter EX1 3PB, UK. [2] NCAS-Climate, University of Reading, Reading RG6 6BB, UK. [3] Los Alamos National Laboratory, Los Alamos, New Mexico 87545, USA. Correspondence and requests for materials should be addressed to P.G. (email: peter.good@metoffice.gov.uk).

According to ensemble mean projections from the fifth coupled model intercomparison project (CMIP5), global mean temperatures could reach at least 2 K above pre-industrial levels[1], even with significant mitigation action (here, pre-industrial refers to the CMIP5 pre-industrial control simulations). This magnitude of change has been taken as a target to avoid potentially dangerous anthropogenic interference with the climate system. This level may be reached by mid-century if high anthropogenic emissions continue (as in the RCP8.5 scenario), or near the end of the century under relatively strong mitigation conditions (that is, for emissions between those of the RCP2.6 and RCP4.5 scenarios). If high emissions persist, a second 2 K of global warming could occur by the end of the century.

Stakeholders need information, not just about global mean warming but on likely regional-scale climate changes—especially for precipitation[2]. Some require information on change over the coming decades (for example, to inform adaptation policy), while for others the potential benefits of mitigation action later in the century is of more interest (that is, the difference in climate change between high- and low-emission scenarios). It is important to quantify and understand likely climate changes under these distinct conditions, but they are not always examined separately.

A significant amount of current work assumes, either implicitly or explicitly, that in any given climate model each K of global warming produces roughly the same changes in regional climate. This assumption is implied in some studies of physical mechanisms, where it is common[3] to examine the climate changes under a high-forcing scenario (for example, RCP8.5 at the end of the century, or the suite of high-forcing idealized experiments, designed for studying mechanisms, in CMIP5). If the results from this kind of study are to be applied to lower-forcing scenarios or periods of stakeholder interest, one must assume that the balance of mechanisms driving regional climate change is roughly constant—or, equivalently, that the spatial patterns of precipitation change are roughly constant, and simply scale with global mean warming. The assumption of a fixed spatial pattern of climate change per K of global warming is also explicit in some policy tools (usually called pattern scaling[4–7]).

The above assumption is known not to apply exactly[4,8–10], and physical reasons exist to challenge it, but have yet to be thoroughly investigated. These reasons fall into three categories: a changing balance of different forcing agents (which cause different patterns of climate response[11]), and linear and nonlinear mechanisms of climate response[12].

The most obvious reason for precipitation patterns to differ between different scenarios or time periods is a changing balance of forcings. Different forcings in general produce different patterns of climate response, so if their balance changes (even with a constant total), the climate patterns will alter[9]. Anthropogenic aerosol, in particular, produces different responses than greenhouse gases[11,13,14], and its temporal evolution, peaking near the end of the twentieth century, is expected to be quite different from that of greenhouse gases under the CMIP5 RCP scenarios[15,16]. Land-use and land-cover change could also have distinct effects on climate change[17]. Even for a constant balance of forcings, however, climate change patterns may differ between different scenarios or periods, due to linear and nonlinear mechanisms[12].

Linear mechanisms are here considered to be those consistent with linear system theory—that is, where doubling the forcing everywhere along a scenario doubles the response throughout (as assumed by linear response function models[18–20]). Even in a linear system, climate change response patterns may evolve due to different timescales of response[21,22]. For example, precipitation has both fast responses to forcing (via reduced tropospheric radiative cooling and rapid land surface warming) and slower indirect responses (via sea-surface temperature warming)[23,24]. Patterns of surface warming (and hence precipitation change[3]) can also evolve over time (for example, warming over the Southern Ocean lags global mean warming[21]). These effects, therefore, are sensitive to the forcing history. If forcings have been changing rapidly (such as under RCP8.5), then fast responses are relatively important, as they adjust quickly to the recent forcing changes. If, on the other hand, forcings have been stabilised for some time (such as under RCP2.6 or RCP4.5 by the end of the century), slower responses have had more time to develop, so play a relatively larger role. Linear mechanisms can alter both global mean precipitation and global climate feedbacks significantly[25–27], but have not been investigated thoroughly for regional precipitation[28].

Nonlinear mechanisms are those inconsistent with linear system theory. These may include state-dependent feedbacks, such as the sea-ice albedo feedback (which vanishes for large or zero sea-ice cover[29]). Nonlinear mechanisms can cause climate patterns to differ at different levels of forcing[12]. For example, if an equivalent of RCP8.5 was run with double the forcing, linear mechanisms would show exactly double the response compared with the standard RCP8.5, but nonlinear mechanisms would not. Nonlinear mechanisms have been demonstrated in a few models for very high-forcing levels[30], or under idealized $CO_2$-forced experiments, for global and regional-scale precipitation[31], warming[12,32–34] and ocean heat uptake[35]. In one model study using idealized experiments[36], nonlinear precipitation change over tropical oceans was associated with interactions between pairs of approximately linear mechanisms (for example, simultaneous moisture increases and circulation shifts). Nonlinear behaviour of the Indian Summer Monsoon associated with the positive moisture advection feedback has also been proposed[37].

However, these studies do not investigate the implications of linear or nonlinear mechanisms of pattern change for policy-relevant scenario projections. Both the relative importance of precipitation pattern change and the relative roles of different mechanisms are unclear. In particular, no information is available on their role in climate model uncertainty in regional-scale precipitation change.

Here we address these gaps directly for CMIP5 scenario projections from 25 climate models. We focus on three different intervals in the CMIP5 simulations that each show differences in global, ensemble mean temperature of close to 2 K. Two of these are fast and slow paths to 2 K above pre-industrial conditions; the third is an interval from 2 to 4 K above pre-industrial (in this case, the difference between a high-forcing and a mitigation scenario at the end of the century). For each of these 2 K intervals, we examine the regional-scale differences in precipitation that accompany the 2 K difference in global, ensemble mean temperature. It is found that, although the two routes to a first 2 K give very similar precipitation changes, a second 2 K produces quite a different response, consistent with a significant influence from nonlinear physical mechanisms.

## Results

### 2 K intervals and the roles of linear and nonlinear mechanisms.
We examine change relative to pre-industrial conditions, rather than relative to a recent historical period. This is done because it simplifies the separation of different physical mechanisms, by reducing pattern changes due to a changing balance of aerosol forcing (which has seen more attention elsewhere[9,14]). In all CMIP5 scenarios, the change in aerosol forcing relative to pre-industrial levels is small by the end of the century, but this is not true for changes relative to recent historical periods.

The historical climate change, from pre-industrial to the recent past, may be considered to be a separate problem that may be tackled using different information, such as real-world observations.

We first study two routes to a first 2 K of global mean warming relative to pre-industrial conditions. This is reached, in the CMIP5 ensemble mean, by mid-century under the reference scenario RCP8.5. This, the fastest route to a first 2 K in CMIP5 projections, is denoted 2 K (Fast).

An alternative, mitigation route to a first 2 K is obtained by averaging the RCP2.6 and RCP4.5 scenarios: the ensemble mean warming is slightly <2 K above pre-industrial levels under RCP2.6, and slightly >2 K under RCP4.5. Taking this mean reduces noise from internal variability and produces a scenario that reaches 2 K in the ensemble mean by the end of the century. This route is denoted 2 K (Mit). Potential issues with averaging RCP2.6 and RCP4.5 are discussed in the Supplementary Note 1.

These two routes to a first 2 K have different warming and forcing histories (compare yellow and blue lines in Fig. 1). As a result, they may show different precipitation patterns due to linear mechanisms (that is, from different timescales of response). Further pattern differences may arise because the mid-century aerosol burden in 2 K (Fast) is higher than that at the end of the century—in 2 K (Mit).

In addition, we study a second 2 K interval, defined as the difference between 2 K and 4 K above pre-industrial levels. This is quantified by the difference between RCP8.5 and our averaged mitigation scenario (the mean of RCP2.6 and RCP4.5), at the end of the century. This 2 K interval can be interpreted as quantifying the climate benefits of mitigation action, and is directly relevant to the debate on climate mitigation. It is denoted 4–2 K (Mit). The forcing history in 4–2 K (Mit) is similar to that in 2 K (Fast)—see red and orange lines in Fig. 1. 4 K–2 K (Mit) should be relatively unaffected by aerosol forcing.

We include results for all CMIP5 models which simulate all of RCP2.6, RCP4.5 and RCP8.5 (25 models; Supplementary Table 1). The same time periods are taken from each climate model, to ensure that they each experience the same forcing history. Hence, individual models may show global mean warming greater or <2 K. Specific averaging periods are given in Methods. We analyse ensemble mean patterns first, then explore the inter-model differences.

**Ensemble means**. For the CMIP5 ensemble mean, the two routes to a first 2 K give very similar patterns of regional precipitation change (Fig. 2a–c). The difference between the two (Fig. 2c, plotted with half the scale of Fig. 2a,b) is small everywhere, except over part of China, which shows a locally large effect attributable to the higher aerosol burden in this region in the middle than at the end of the century[15]: increased aerosol suppresses precipitation in this region[9,14]. Apart from this local effect, it appears that the ensemble mean response is relatively insensitive to how a first 2 K of global ensemble mean warming is achieved. This implies that linear mechanisms (associated with different timescales of response) do not cause a large difference in regional-scale precipitation change patterns under these scenarios, although fast precipitation responses[24] to $CO_2$ may still contribute to the pattern seen in Fig. 2a,b. Away from China, the small differences in Fig. 2c are consistent, in location and approximate magnitude (Supplementary Note 2) with linear mechanisms associated with fast responses to radiative forcing[24]. Nonlinear mechanisms do not contribute significantly to the pattern in Fig. 2c, because the forcing and global mean temperature is similar for the two 2 K intervals.

In contrast, the end-of-century response to a second 2 K, 4 K–2 K (Mit), is rather more distinct (Fig. 2d,e). Figure 2e shows the difference in response between this second 2 K of warming, and the first 2 K under mitigation. These differences are much larger than those shown in Fig. 2c. Positive values are more common at low latitudes and negative values at mid-latitudes, including southern South America. This is not a small effect: over large areas of the land surface the responses for these two 2 K intervals differ by a factor of 1.5 or more (values >3/2 or <2/3 in Fig. 2f).

The effects of linear mechanisms on these results (that is, on Fig. 2e,f) should be relatively small. The forcing history for 4 K–2 K (Mit) is similar to that for the first 2 K under RCP8.5 (compare red and orange lines in Fig. 1b,c). Therefore, the contribution of linear mechanisms to Fig. 2e should be similar to their contribution to Fig. 2c. Figure 2c shows this to be a relatively small effect. In other words, if only linear mechanisms were important, Fig. 2c,e would be similar. Aerosol effects may largely be excluded because aerosol forcing is small relative to pre-industrial levels in all RCPs at the end of the century.

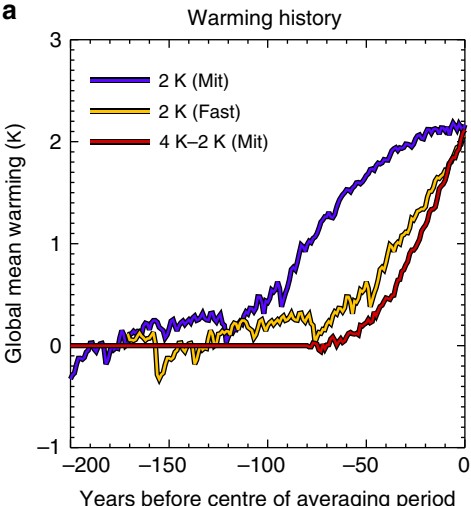

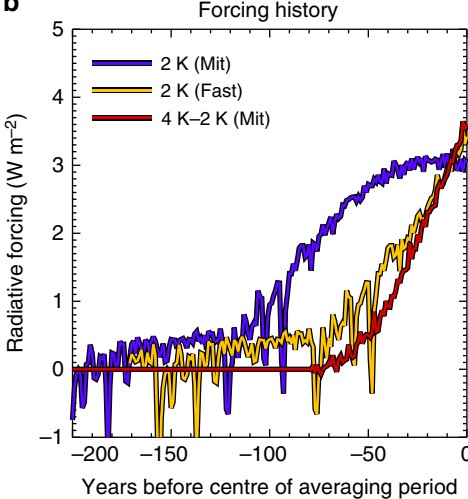

**Figure 1 | Warming and forcing history of the three 2 K warming intervals.** (**a**) Ensemble mean global warming before the centre of each time period of interest (that is, before year 2040 for 2 K (Fast) and 2085 for the other two) for 2 K (Mit); blue, 2 K (Fast); brown/yellow, 4 K–2 K (Mit); red. (**b**) The corresponding radiative forcing history (ensemble mean, calculated using the Forster and Taylor method[41]). Results are shown for a subset of 16 models (Supplementary Table 1) for which the forcing was available (sufficient for illustrating scenario forcing histories).

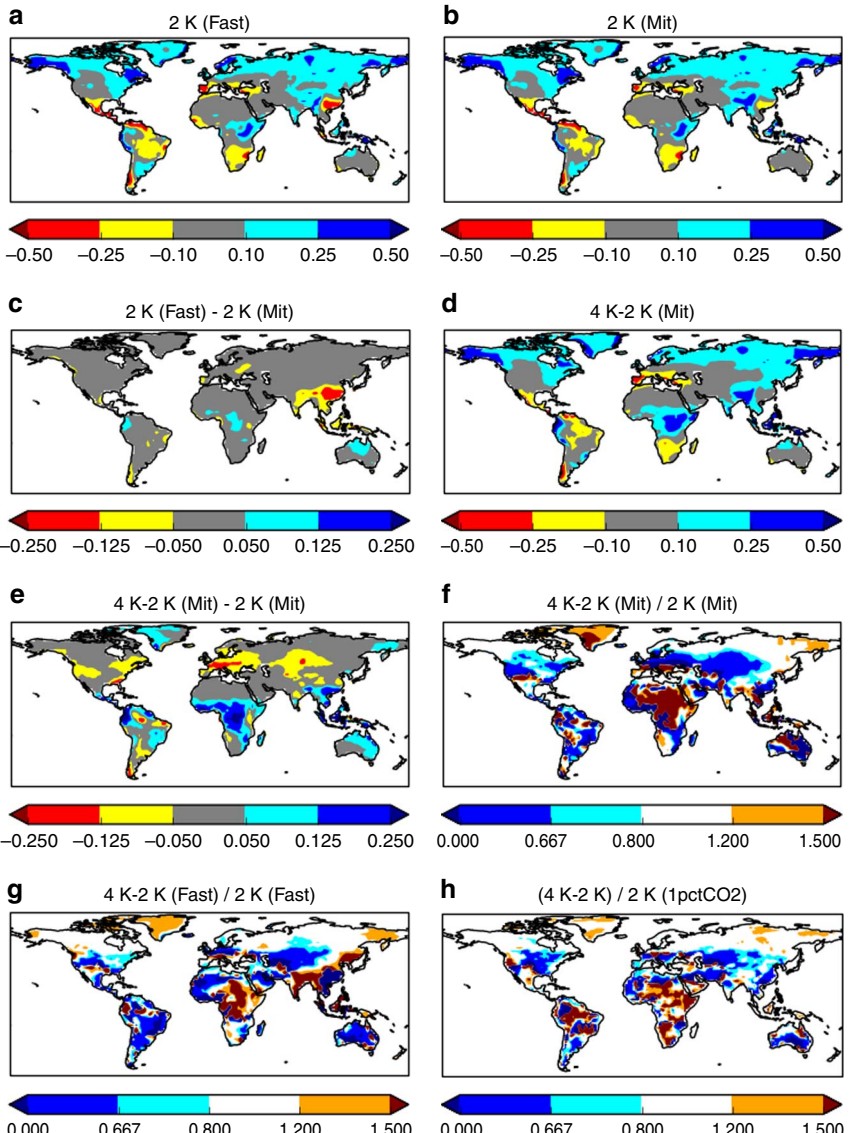

**Figure 2 | Ensemble mean precipitation differences.** Results (in mm per day) are shown for change along (**a**) the fast (high-forcing) and (**b**) the mitigation routes to a first 2 K; and (**c**) their difference (scale for **c** is half that of **a,b**). (**d**) Differences for the second 2 K; (**e**) the second 2 K minus the first 2 K under mitigation conditions (note different scale in **c**). (**f**) The second 2 K divided by the first 2 K under mitigation conditions. (**g**) as (**f**) but using data from RCP8.5 only. (**h**) as (**f**) but for the 1pctCO2 experiment.

Figure 2g also shows ratios between precipitation change under a second and a first 2 K of warming (as Fig. 2f), but just for the RCP8.5 scenario (for the second 2 K under RCP8.5, divided by the first 2 K under the same scenario). Away from China, Fig. 2g is similar to Fig. 2f, with slightly smaller magnitudes, due to a different effect of linear mechanisms (arising from different forcing histories).

We find similar pattern differences in the idealized 1pctCO₂ experiment, which is forced by $CO_2$ changes only (Fig. 2h). Combined with the above results, this implies that nonlinear responses to greenhouse gas forcing are the primary influence behind Fig. 2e–g, although other forcings such as aerosol and land-use are also likely to be important locally. Underlying nonlinear mechanisms are likely to include interactions between roughly linear processes[36], or nonlinear evapotranspiration effects[12] or rainband shifts[30].

The above results are presented in units of mm per day. The same broad conclusions are found if results are given as percentages of the local pre-industrial climate mean

(Supplementary Figs 2 and 3). In particular, the ratios (compare Fig. 2f and Supplementary Fig. 3c) are insensitive to whether precipitation change is expressed in absolute or relative terms.

Some large model uncertainties exist, however. Supplementary Fig. 4 shows estimates of the signal/noise ratio in the ensemble mean, corresponding to Fig. 2a–e. In most cases, the ensemble mean patterns coloured in Fig. 2a–e are relatively robust, with signal/noise ratios > 2, although individual models may disagree on the sign of change. However, signal/noise ratios are weaker (Supplementary Fig. 4e) for tropical regions of Fig. 2e: so, while the differences in rainfall response between a first and second 2 K can be relatively large in the tropics (Fig. 2e,f), model uncertainty in this behaviour is also large in this region. The next section explores how this alters the inter-model differences between the different 2 K intervals.

**Inter-model differences.** To reduce uncertainty of climate projections, it is important to understand physically the differences between different model simulations. We now compare the

inter-model differences for different 2 K intervals. Underlying this is the question: is the balance of physical mechanisms driving the inter-model differences the same for each 2 K interval? If the answer is yes, then a model that exhibits strong drying at a given location for one 2 K interval should also dry strongly for other 2 K intervals. Therefore, if, for this location, we plot the precipitation response for one 2 K interval against the response for another (one point per model), a strong correlation should be seen.

This is indeed the case when the two routes to a first 2 K are compared over the Western Amazon (Fig. 3a). However, a rather weaker correlation is found (Fig. 3b) when 4 K–2 K (Mit) is compared with 2 K (Fast). That is, inter-model differences in precipitation change over this region look similar for the two routes to a first 2 K, but different for the second 2 K. Hence, the main physical reasons for the model differences over the Western Amazon may be different for warming from 2 K to 4 K than for a first 2 K. A similar picture is seen over Western Africa—one of the most densely populated areas of Africa (Fig. 3c,d). For this

region, both the inter-model differences and the ensemble mean are quite different for 4 K–2 K (Mit) than for a first 2 K. Both nonlinear mechanisms and land-use change could influence these results, but a strongly nonlinear precipitation response to $CO_2$ forcing over this region has been shown in the HadCM3 climate model[31]. In contrast, inter-model differences are rather similar for all three 2 K intervals over East Africa (Fig. 3e,f). This is consistent with the hypothesis that inter-model disagreement in this region is caused by similar physical mechanisms for all three 2 K intervals. These regions were chosen just to illustrate the range of behaviours. Other regions also show differences in inter-model spread between different 2 K intervals.

The regional mean results in Fig. 3 are broadly typical of other regions, especially in the tropics (Fig. 4a–c). Our primary result here is in Fig. 4c: over most, but not all regions, the two routes to a first 2 K are more strongly correlated with each other than with the second 2 K response. The correlation coefficients often differ by a factor of 2 or more (Fig. 4c). This suggests that in many

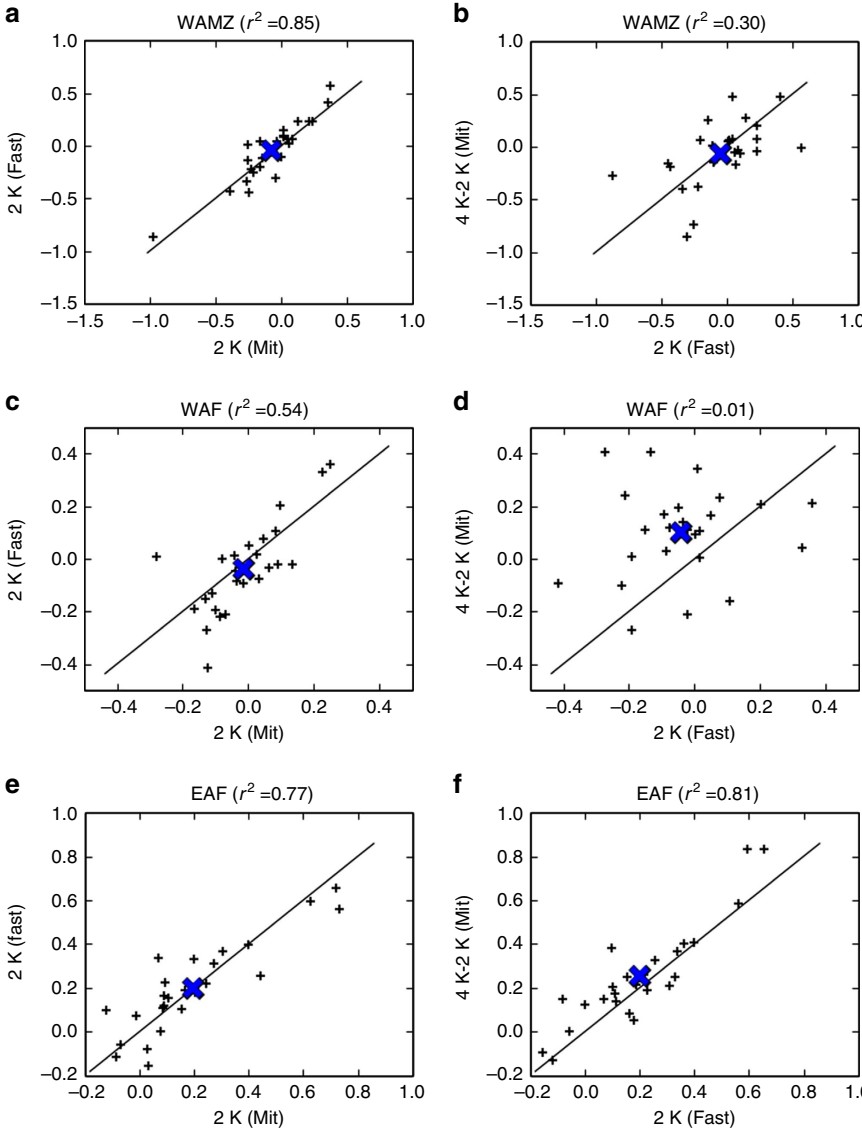

**Figure 3 | Regional comparison of inter-model differences for different 2 K intervals.** Results show precipitation (mm per day) for three different 2 K intervals, over three regions. All panels: one small cross per model; large blue cross shows ensemble mean. Left (**a**,**c**,**e**): comparing the two routes to a first 2K. Right (**b**,**d**,**f**): the second 2 K versus 2 K (Fast). Analysis is chosen so that internal variability has the same influence on the left and right columns, and so that variability in each pair of variables is independent (Methods). Region definitions: Western Amazon (WAMZ[12]): 12 S–3 N, 72–60 W; Western Africa (WAF): 0–12 N, 20 W–15 E; Eastern Africa (EAF): 3 S–12 N, 40–52 E (regions are marked in Fig. 4c).

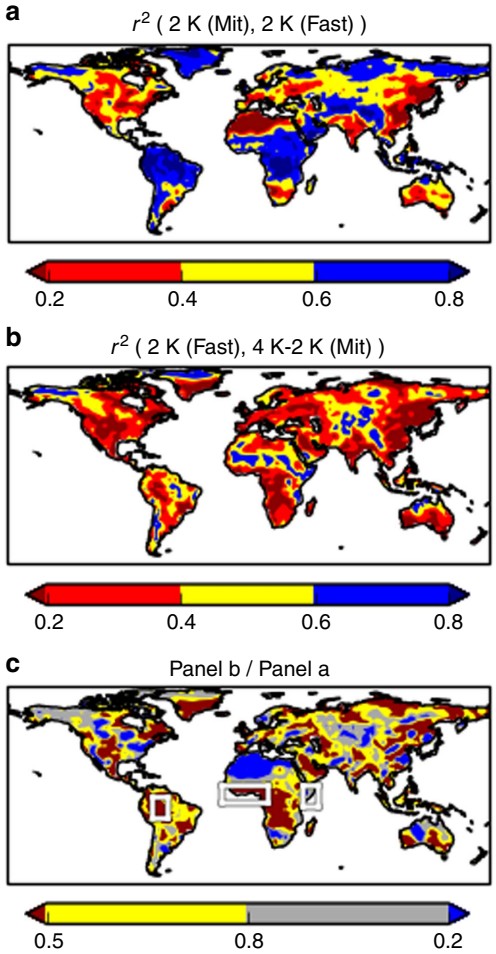

**Figure 4 | Global comparison of inter-model differences for three 2 K intervals.** Maps show comparisons between (**a**), the two routes to a first 2 K (as left column of Fig. 3); and (**b**), the second 2 K and 2 K (Fast)—as right column of Fig. 3. Values are Pearson correlation coefficients (squared, for variance explained) between the vector of 25 model results for one 2 K interval and the equivalent vector for the second 2 K interval. (**c**) panel **b**/panel **a**. Regions marked in white in **c** correspond to those used in Fig. 3.

regions the balance of mechanisms driving model differences for a second 2 K may be different than that for a first 2 K, but further investigation of specific mechanisms is required.

This result is important for efforts to understand and reduce uncertainty in regional precipitation change: it implies that it is inaccurate to think of the balance of physical mechanisms driving this uncertainty to be constant across the range of conditions explored by the CMIP5 scenarios. For example, studies using observable constraints based on the present-day variability[38] may have limited relevance to quantifying the benefits of climate mitigation action.

The differences between the two routes to a first 2 K are uncorrelated (across models) with the differences between a first 2 K and a second 2 K. We show this by calculating, at each location, two differences for each model: (2 K (Mit) − 2 K (Fast)) and (4 K–2 K (Mit) − 2 K (Fast)). At each location, we calculate the correlation between these two differences across the 25 models. Over 99% of the land area, $r^2$ is <0.2. This result is consistent with the idea that different mechanisms are highlighted by these two differences: primarily linear mechanisms and aerosol when comparing the two routes to a first 2 K; and nonlinear mechanisms and possibly land-use for the second 2 K versus a first 2 K.

At mid-latitudes, correlation coefficients are generally lower, even in Fig. 4a. This is probably an artefact of internal variability: the ratio between internal variability and inter-model spread in precipitation change is larger here than in the tropics[39]. For this reason, we focus on the ratios in Fig. 4c rather than the absolute values in Fig. 4a,b.

Although the spatial patterns in Fig. 4 may give useful clues as to which regions to prioritize for further investigation, caution is required in interpreting these patterns. The correlation coefficient at any given location has some statistical uncertainty due to the finite number of models. For our purposes, it is sufficient to note that the values in Fig. 4c are generally much <1.

Using the same arguments as for the ensemble mean, these results together suggest that uncertain nonlinear mechanisms, combined with land-use effects in some regions, are the primary factor altering the balance of uncertain mechanisms between the first and second 2 K, although linear mechanisms will also play a role.

## Discussion
Our results have implications most directly for efforts to understand climate change mechanisms (in particular, for understanding mechanisms behind climate model disagreement), but potentially also for quantitative policy advice tools and for what the observable climate can tell us about future change. We find that, for precipitation, the task of quantifying and understanding the benefits of mitigation—what climate impacts can be avoided by taking mitigation action—is a somewhat different problem than understanding responses to climate change under low-forcing conditions (that is, over the next few decades, or under strong mitigation conditions such as RCP2.6 or RCP4.5). On the other hand, patterns of precipitation change are relatively insensitive to how a first 2 K of global ensemble mean warming is reached, relative to pre-industrial conditions. This supports the approach of quantifying some impacts under +2 K and +4 K conditions, as in work towards the next UK Climate Change Risk Assessment[40].

Evidence for strong nonlinearities in some regions[31,36] points to a challenge in better understanding the underlying physical mechanisms, with implications for how observations are used to improve parameterisations of cloud, convection and other rainfall processes. There may be some links to the nonlinear mechanisms identified previously[12] for surface warming. In particular, effects related to change in the Bowen ratio over the west Amazon and tropical Africa[12] could plausibly play a role in results in similar regions in Fig. 4c. Behaviour of seasonal mean rainfall and other climate variables will be investigated in future. There is an opportunity, in the forthcoming CMIP6 for comparing responses to abrupt2 × $CO_2$ (part of CFMIP) and the conventional abrupt4 × $CO_2$. It is hoped that this experiment will see widespread engagement of the climate modelling community, alongside experiments targeting other forcings, such as aerosol and land-use change.

There are also potential implications for quantitative policy tools using pattern scaling[6,7], which by definition assume the patterns of climate change for each K of global warming are approximately constant. Some ensemble mean scenario dependence of precipitation change patterns has previously been shown[8], notably in the RCP2.6 extension to year 2300. Some differences in spatial patterns stand out more clearly in the current study due to the use of longer-term means, and because we study the second 2 K of warming independently of the first 2 K (the second 2 K is relevant to mitigation advice). Relative changes in precipitation patterns per K of global warming shown here are significantly larger than for temperature[12]. Some techniques

already exist to improve pattern scaling. Attention has focused on aerosol forcing[9], and on linear mechanisms (the latter, by adding greenhouse gas forcing[9] or land-sea contrast[10] as additional predictors alongside global warming, or by using response function methods[12]). A time-shifting method also exists[10] that could reduce error from some nonlinear mechanisms. This method could perform better if effects from non-greenhouse gas forcings and linear mechanisms are first removed (for example, Fig. 2e of ref. 10 shows errors over China that are consistent with aerosol forcing). The most appropriate method will be application-dependent, according to the region, scenario and time-period studied and on the magnitude of internal variability, which depends on the time-averaging period chosen.

Figure 4b implies a possible limit to which observable past precipitation changes could constrain uncertainty about change under further global warming. According to this ensemble, even if the regional precipitation change for a first 2 K of global warming was known precisely, this may not greatly reduce uncertainty about change under a second 2 K. For example, over the Western Amazon, the first 2 K response explains only about a third of the inter-model variance for the second 2 K (Fig. 3b). Similarly, observational constraints based on the present-day variability[38] may be biased for precipitation change, especially for quantifying the benefits of mitigation.

## Methods

**Model data and 2 K intervals.** Results show annual mean precipitation from 25 CMIP5 models (listed in Supplementary Table 1). A single initial value ensemble member is used from each model, for each of four simulations: the pre-industrial control, and the RCP2.6, RCP4.5 and RCP8.5 projections. Results from all models were regridded to a common 2.5° grid. Data are converted to anomalies with respect to the pre-industrial control mean before analysis.

**2 K intervals and choice of averaging periods.** The 2 K intervals are as follows: 2 K (Fast): change by mid-century under RCP8.5; 2 K (Mit): change by the end of the century under the averaged mitigation scenario (the mean of RCP2.6 and RCP4.5); 4 K–2 K (Mit): the difference between RCP8.5 and the averaged mitigation scenario, at the end of the century.

End-of-century results, for 2 K (Mit) and 4 K–2 K (Mit) are all averaged over the 30-year period 2071–2100. Thus, the averaged mitigation scenario (mean of RCP2.6 and RCP4.5) involves a mean over 60 years of data. This 60-year mean has relatively low noise from internal variability. This is important for this period in particular as it is used in both 2 K (Mit) and 4 K–2 K (Mit). So, in principle, noise from internal variability could disproportionately affect their difference (shown in Fig. 2e). The small differences seen in Fig. 2c suggest that internal variability is not large. For 2 K (Fast), RCP8.5 is averaged over the 60-year period, 2011–2070. For 2 K (Mit) and 2 K (Fast), anomalies are taken with respect to the 30-year pre-industrial period 1861–1890 (an additional historical period is used in Figs 3 and 4—see below). This means that internal variability has a similar magnitude in each of the three 2 K intervals (each is a difference between a 30-year mean and a 60-year mean).

For a subset of nine models (those that also include a second initial value ensemble member for RCP8.5), a similar analysis was performed where 60-year means were used for the pre-industrial and end-of-century RCP8.5 periods (that is, averaging the two initial value ensemble members for the latter). This produced similar results to those shown in the main paper.

**Internal variability.** The analysis in Figs 3 and 4 is constrained by two issues.

First, no-longer than 30-year means are possible for the 4 K climate state (that is, for RCP8.5, end of century). To ensure that the magnitude of internal variability is consistent, each 2 K interval is evaluated as a difference between a 30-year mean and a 60-year mean (see 'Scenario averaging periods' above).

Second, we ensure that the internal variability in each pair of 2 K intervals being correlated is statistically independent. This is necessary to avoid bias in the correlation between these variables. Therefore, when comparing the two routes to a first 2 K (left column of Figs 3 and 4a), different historical periods are used for 2 K (Fast) and 2K (Mit) (1861-1890 and 1911-1940 respectively). For the same reason (ensuring independent variability), 4 K–2 K (Mit) is compared with 2 K (Fast) in the right column of Figs 3 and 4b. Swapping the two historical periods in this analysis has no effect on the conclusions drawn from Figs 3 and 4a: only minor changes in the pattern in Fig. 4a are seen.

**Data availability.** The CMIP5 data used here are publically available from the CMIP5 web portal at http://cmip-pcmdi.llnl.gov/cmip5/.

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

## Acknowledgements

This work was supported by the Joint UK DECC/Defra Met Office Hadley Centre Climate Programme (GA01101) and by the Office of Science (BER), US Department of Energy. E.H. was supported by a NERC Advanced Fellowship and the UK National Centre for Atmospheric Science. B.B.B.B. was also partially funded by the DfID/NERC HyCRISTAL (NE/M02038X/1) and AMMA-2050 (NE/M019977/1) projects. We acknowledge the World Climate Research Programme's Working Group on Coupled Modelling, which is responsible for CMIP, and we thank the climate modelling groups (listed in Supplementary Table 1 of this paper) for producing and making available their model output. For CMIP, the US Department of Energy's Program for Climate Model Diagnosis and Intercomparison provides coordinating support and led development of software infrastructure in partnership with the Global Organization for Earth System Science Portals. Some helpful discussion with Hugo Lambert improved the manuscript.

## Author contributions

P.G. conceived the study and wrote the paper. All authors contributed to the scientific interpretation and the paper.

## Additional information

**Competing financial interests:** The authors declare no competing financial interests.

