## [Peer Review File · Nature Communications]

Reviewers' comments:

Reviewer #1 (Remarks to the Author):

The paper describes a novel study which shows that regional precipitation responds differently to the second 2K of global warming compared to the first 2K. This is a result with important implications for adaptation and mitigation policy. The analysis in the paper is sound and supports the papers conclusions, therefore, I recommend publishing the manuscripts after a few minor revisions have been completed.

The authors could aid readers by clarifying their methods better in a few key places. For example, in line 130, they could define here what they mean by precipitation changes (i.e. compared to what) for each of the 3 intervals. Similarly in Figure 2 caption.

Figure 2, a statistical test (e.g. student t-test) could show where they ensemble means of the (a) and (b) and (d) and (b) are significantly different given model spread (i.e. internal variability and model differences).

Line 250 onwards/Figure 3: Explain why these regions where chosen as other regions seem to show as large differences in Figure 2(c) and (e), e.g. East Asia. It would also help to plot a box around the chosen regions on Figure 2 or 4.

Line 334: "not NOT?"

Reviewer #2 (Remarks to the Author):

1. What are the main claims of the paper and how significant are they?

The paper shows a non-linear response in precipitation change for higher-end forcing of climate change, compared to lower forcing (or mitigation). These are significant because there has largely been an assumption of linearity in response across different likely emissions scenarios for the 21st century - that is for each 1K in global T there is a proportional, consistent spatial pattern of change in climate variables such a temperature and precipitation

2. Who will be interested in reading the paper, and why?

The paper will be of interest to those interested in using climate model projections for policy, and for adaptation planning, because, I think for the first time it shows that are shifts away from a linear response to forcing with regard to precipitation changes as one goes to high-end forcing. It also opens up a series of research questions around the mechanisms behind this non-linear response that the authors allude to, but are not able to dig into within the scope of this paper.

3. Is the paper likely to be one of the five most significant papers published in the discipline this year?

It is likely to be highly significant within the narrow sub-area of climate model evaluation, analysis, and policy implications; within the broader climate change domain, this is likely to be of interest, but not one of the top 5 papers (across IPCC WG1, 2, 3 for instance).

4. How does the paper stand out from others in its field?

The big difference is in comparing precipitation response when ensemble mean of models reach 2 and 4K of warming, and then the difference between first and second 2K of the 4K warming simulations. It is similar in some ways to a previous paper looking at linearity in temperature responses, by the same lead author (Good et al, 2015), but takes a different approach by looking at 4k-2k, and analysing precipitation.

5. Are the claims novel? If not, which published papers compromise novelty?

I think that the claims are novel - I am not aware of any similar work that has separated out patterns of change across an ensemble and explicitly looking at fast versus slow warming to 2k, and then difference between 2k and 4k.

6. Are the claims convincing? If not, what further evidence is needed?

The claims around the similarity in precipitation response between slow/fast warming to 2k and difference in response between first and second 2k of warming are convincing. However the statements around the underlying non-linearity in mechanisms are too strongly stated in my view - I would suggest modification to say that the results for precip between first and second 2k of warming are consistent with non-linearity in process, and then to speculate on what these processes might be.

7. Are there other experiments or work that would strengthen the paper further?

I think there is enough in this paper for a communication - it sets an agenda for further research looking to see if the same holds on a seasonal basis and to investigate specific mechanisms of non-linearity in future work by this group and others.

8. How much would further work improve it, and how difficult would this be? Would it take a long time?

N/A

9. Are the claims appropriately discussed in the context of previous literature?

I made some comments in the manuscript that could add to discussion - please see annotated manuscript.

10. Minor Comments - please see annotated manuscript.

Reviewer #3 (Remarks to the Author):

Review of Good et al, Large differences in regional precipitation change between a first and second 2K of global warming, submitted to Nature Communications

This paper describes the projected rainfall changes for a 2K increase in global temperatures over pre-industrial via two scenario pathways, finding the pathway matters little to the precipitation response. It then contrasts these to a second 2K increase comparing the late 21st Century warming from RCP8.5 to that of the average of RCP2.6+RCP4.5, finding substantial non-linear responses in precipitation for this second 2K. It is a well-written and carefully crafted study using a novel approach with interesting results and I recommend publication.

My only major comment is that while the paper provides much background into previous literature on non-linear rainfall responses, it has very little discussion of its results in direct comparison to recent studies assessing the robustness of pattern scaling techniques. In particular, a number of recent studies listed below have looked at the robustness of pattern scaling across policy relevant (RCP) scenarios and highlighted regions of model disagreement for precipitation. The results here need to be placed in better context with those studies.

Frieler, K., M. Meinshausen, M. Mengel, N. Braun, and W. Hare (2012), A scaling approach to probabilistic assessment of regional climate change, *J. Clim.*, 25(9), 3117-3144, doi:10.1175/JCLI-D-11-00199.1.

Tebaldi, C., and J. M. Arblaster (2014), Pattern scaling: Its strengths and limitations, and an update on the latest model simulations, *Clim. Change*, 122(3), 459-471, doi:10.1007/s10584-013-1032-9.

Herger, N., B. M. Sanderson, and R. Knutti (2015), Improved pattern scaling approaches for the

use in climate impact studies. *Geophys. Res. Lett.*, 42, 3486-3494. doi: 10.1002/2015GL063569.

Also, how does the presumed larger spread in global temperatures for the second 2K (since the larger values of warming to 4K will presumably lead to more model divergence around the ensemble mean) impact the results in Fig 3 and 4.

Minor comments:

Figure 1: 4K-2K(Mit) would be more understandable if relabelled to 4K(Fast)-2K(Mit)

Figure 3: the caption needs improving to note that the precipitation response is being plotted and units provided

Line 130-142: suggest moving these details after first introducing the experiments

Line 132: referring to Methods here is confusing as the experiments have yet to be introduced (and this is assumed knowledge in the Methods)

Line 137: but pre-industrial conditions vary widely across models, particularly the magnitude of aerosol forcing used

Line 184: the main region of difference is China, not South East Asia (the latter being the countries to the south of China)

Response to Reviews

Reviewer #1 (Remarks to the Author):

The paper describes a novel study which shows that regional precipitation responds differently to the second 2K of global warming compared to the first 2K. This is a result with important implications for adaptation and mitigation policy. The analysis in the paper is sound and supports the papers conclusions, therefore, I recommend publishing the manuscripts after a few minor revisions have been completed.

Many thanks for these comments, which have led to a better paper.

The authors could aid readers by clarifying their methods better in a few key places. For example, in line 130, they could define here what they mean by precipitation changes (i.e. compared to what) for each of the 3 intervals. Similarly in Figure 2 caption.

Yes, perhaps 'change' is not the best word, given that one of our 2K intervals is a difference between scenarios. We attempt to clarify this as follows:

- At the start of the Results, we now write, "Here we focus on three different intervals in the CMIP5 simulations, that each show differences in global, ensemble mean temperature of close to 2K. Two of these are fast and slow paths to 2K above pre-industrial conditions; the third is an interval from 2K to 4K above pre-industrial (in this case, the difference between a high-forcing and a mitigation scenario at the end of the century). For each of these 2K intervals, we examine the regional-scale differences in precipitation that accompany the 2K difference in global, ensemble mean temperature."
- The Figure 2 caption now starts: "Ensemble mean precipitation differences (mm/day) for change along **a** the fast (high-forcing) and **b** the mitigation routes to a first 2K; and **c**, their difference (scale for panel **c** is half that of **a,b**). **d** shows differences for the second 2K"

Figure 2, a statistical test (e.g. student t-test) could show where they ensemble means of the (a) and (b) and (d) and (b) are significantly different given model spread (i.e. internal variability and model differences).

In the spirit of the Reviewer's suggestion, we include a new supplementary figure 4, referenced at the end of the section on the Ensemble mean. This plots the ratio $|\text{ensemble mean}| / (\text{standard deviation in the ensemble mean})$ corresponding to Figures 2a-e. This is useful as it shows that model uncertainty in the tropics is relatively large for Figure 2e, consistent with the following section on inter-model differences. This is described in a new paragraph (end of section on ensemble means):

“Large model uncertainties exist, however. Supplementary Figure 4 shows estimates of the signal/noise ratio in the ensemble mean, corresponding to Figures 2a-e. In most cases, the ensemble mean patterns coloured in Figures 2a-e are more than 2 standard deviations from zero, suggesting that similar results for the ensemble mean may be found in the next generation of climate models (although individual models may disagree on the sign of change). However, signal/noise ratios are weaker (Supplementary Figure 4e) for tropical regions of Figure 2e: so, while the differences in rainfall response between a first and second 2K can be relatively large in the tropics (Figures 2e,f), model uncertainty in this behaviour is also large in this region. The next section explores how this alters inter-model differences between the different 2K intervals.”

Line 250 onwards/Figure 3: Explain why these regions were chosen as other regions seem to show as large differences in Figure 2(c) and (e), e.g. East Asia. It would also help to plot a box around the chosen regions on Figure 2 or 4.

These regions were just chosen as being illustrative. We now state: “These regions were chosen just to illustrate the range of behaviours. Other regions also show differences in inter-model spread between different 2K intervals.”

The regions are now marked in Figure 4 (this is referred to in the caption of Figure 3).

Line 334: "not NOT?"

Fixed – apologies for that strange error!

Reviewer #2 (Remarks to the Author):

Many thanks for the very helpful comments, which have led to an improved manuscript.

Minor Comments (copied from annotated manuscript).

Abstract:

See further in text, but I don't believe you have proven non-linear mechanisms. Adjust to say that results are consistent with what might be expected if non-linear mechanisms are coming into play.

This sentence now starts, "The results are consistent with a significant influence from nonlinear physical mechanisms..."

See comments in text: this part of abstract might need to be changed. (land-use)

Land use is included in these scenarios. We include a reference (see below) citing a relevant paper.

It would be good to expand this a little to give examples of fast and slow responses for precipitation - these will not be known by many of the non-specialists who might be interested in this paper.

We now write, "For example, precipitation has both fast responses to forcing (via reduced tropospheric radiative cooling and rapid land surface warming) and slower indirect responses (via sea-surface temperature warming)". These points are expanded in the references provided.

Results

It is not clear why you do this. In the introduction, you say that the changes in the balance of different types of forcing might be important for regional changes in precipitation, yet now you are aiming to minimise this effect.

Yes, this could be clearer. The manuscript now reads, "We examine change relative to pre-industrial conditions (rather than relative to a recent historical period). This is done because it simplifies the separation of different physical mechanisms, by reducing pattern changes due to a changing balance of aerosol forcing (which has seen more attention elsewhere). In all CMIP5 scenarios, the change in aerosol forcing relative to pre-industrial levels is small by the end of the century (not true for changes relative to recent historical periods)."

It is not clear why you are mentioning this experiment here - you do not seem to have analysed its results? If this is a suggestion for further work, then move to the discussion; if you have actually analysed this experiment, then why not report the results?

Good point, thanks. We have deleted the relevant text, as this experiment (abrupt2xCO₂) is mentioned in the Discussion.

The ensemble mean patterns and the intra-model means could be compared to IPCC AR5 WG1 Figure SPM.8, and similar diagrams; it looks to me like the pattern of lack of consistency of

response across the ensemble for RCP8.5 (areas without hatching) might correspond with the areas where there is a non-linear response between 2K and 4-2K change patterns. So is this analysis starting to tell us something about why we might "trust" high end projections in some regions more than others?

This is an interesting thought that could be addressed properly in future work, but the similarity with these AR5 results does not seem to be close enough for speculating at this stage. The idea of where we might 'trust' high end projections is certainly a target, but we need to understand mechanisms better first.

(old line 248) If you have to add a clarifying sentence, "in other words", it means the previous sentences are not clear - suggest you simply rephrase the entire paragraph to make it read better and more clearly.

We have given this some thought, and realised that what we really mean is 'Therefore', not 'In other words'. (The final sentence of this paragraph is a consequence of the early statements, and is necessary to explain the analysis.) We have changed the text accordingly.

Suggestion for fig 3. Would be nice to see if there are large shifts if the nature of the correlation for individual models - for example is one that is highly +ve correlated for 2k:2k similarly +ve correlated for 4-2k:2k? One way to do this is to use different numbers of symbols for the models; another is to combine the two graphs with a dashed line linking the two correlation values for each model. The latter option would also reduce the space taken by each case, and you could add some different examples from mid-latitudes.

We did some global-scale (land) analysis that addresses this interesting point. Since the reviewer is talking about individual models, we interpret the reviewer's point as: 'Do models that lie off the 1:1 line in the left hand column of Figure 3 also lie off the line in the right hand column (for the same region)? The distance off the line in the left hand column is quantified by the difference: $2K(\text{Mit}) - 2K(\text{Fast})$. The equivalent distance in the right hand column is given by the difference $(4K-2K(\text{Mit}) - 2K(\text{Fast}))$. So we examined whether models that have a large positive value for $2K(\text{Mit}) - 2K(\text{Fast})$ also tend to have a large positive value for $(4K-2K(\text{Mit}) - 2K(\text{Fast}))$. The answer is no. We include a new paragraph to this effect:

"The differences between the two routes to a first 2K are uncorrelated (across models) with the differences between a first 2K and a second 2K. We show this by calculating, at each location, two differences for each model: $(2K(\text{Mit}) - 2K(\text{Fast}))$, and $(4K-2K(\text{Mit}) - 2K(\text{Fast}))$. At each location, we calculate the correlation between these two differences across the 25 models. Over 99% of the land area, r^2 is less than 0.2. This result is consistent with the idea that different mechanisms are highlighted by these two differences (primarily linear mechanisms and aerosol when comparing the two routes to a first 2K; nonlinear mechanisms and possibly land-use for the second 2K versus a first 2K)."

Are you sure that the CMIP5 models include land-use change; this would need to be the case to suggest that LU change is part of the non-linear processes here.

We include (Introduction, in the paragraph beginning “The most obvious reason for precipitation patterns to differ”) a reference to Brovkin et al., which confirms that these simulations do include land-use.

A second point, relating to Amazon, and perhaps other large areas with largely intact natural veg, such as central Africa - do all the CMIP5 models have a dynamic land-surface component? I don't think they do, and so you are missing a potential non-linear feedback as vegetation responds to a wetting/drying over Congo/Amazon.

Certainly not all models include vegetation dynamics. This is an interesting point, but we feel it is too early to suggest in the manuscript that this is a major missing source of nonlinear response (e.g. HadGEM2-ES does have dynamic vegetation, but we don't think vegetation shifts is a big driver of nonlinearity in that model; but it may be in others).

I would be more cautious - either say the graphs are "consistent with the notion that similar mechanisms are at play" (meaning you cannot and have not proved similar mechanisms), or add in a work like "possibly" to the existing sentence.

We now write, “This is consistent with the hypothesis that inter-model disagreement in this region is caused...”

Again, you have not proved this statement - the results are consistent with the hypothesis or notion that different mechanisms might be play between first and second 2ks.

We now write, “This suggests that in many regions the balance of mechanisms driving model differences for a second 2K may be different than that for a first 2K, but further investigation of specific mechanisms is required.”

See previous point on whether land-use change is actually represented in the models. Thinking about it further, if you are suggesting the difference between 2k and 4-2k might partly be due to land use change, these would have to differ in the RCP8.5 scenario and the others - especially over the second half of the 21st century where the big difference between the scenarios occurs. Can you confirm whether there are (i) land-use change scenarios in the RCPs and (ii) that these are different for the different RCPs, and (iii) that these are implemented in the CMIP5 model simulations.

All this is confirmed in the aforementioned paper (Brovkin et al.). N.B. the historical land-use change is also relevant for the first 2K.

Discussion

Include some discussion on the reliability of precipitation-related sub-components of climate models. Precip as a model diagnostic is heavily dependent on an interlinked sequence of parameterisations relating to cloud, convective and large scale precipitation, etc. These are generally at least tuned and parametrised to fit to present day observations. Does the divergence in precip results at 4k also potentially say something about the parameterisations being used outside of their "comfort zone"?

We include a brief mention of the issue of how observations are used to constrain the different sub-processes in paragraph 2 of the Discussion: "...a challenge in better understanding the underlying physical mechanisms, with implications for how observations are used to improve parameterisations of cloud, convection and other rainfall processes." We don't currently understand the mechanisms well enough to say very much here, but certainly the issues raised by the reviewer are important for future work.

Can you compare the results for this paper with that by Good (ref 9) for temperature - are the non-linearities areas biggest difference / non-linearity the same for temp as for precip in this paper?

Yes, this is a good point. In paragraph 2 of the Discussion we now write, "There may be some links to the non-linear mechanisms identified previously for surface warming. In particular, effects related to change in the Bowen ratio over the west Amazon and tropical Africa could plausibly play a role in results in similar regions in Figure 4c."

Would be good to have some discussion on the implications for pattern scaling, as mentioned in the introduction - what do these results mean for the still widely used pattern scaling approaches? The Mitchell paper is one of the few to previously try and look at linearity in scaling from (near) 4k back to 2k patterns, and I seem to remember identified some non-linearities in precip (albeit with only one model, HadCM2 or was it HadCM3?). Do the results here provide a more general confirmation of Mitchell's single model results?

We have expanded the discussion on pattern-scaling (paragraph in Discussion starting, "There are also potential implications for quantitative policy tools"); including reference to some papers that have attempted to improve this technique. We cite the Mitchell paper in the introduction, stating that some changes in precipitation patterns have been previously reported.

Second, statistical downscaling relies on the assumption of linearity, but if there are non-linearities that emerge under 4k relative to 2k, that suggests that SD might start to become less reliable for high-end climate change assessments.

This is an interesting point, but our results do not seem so directly relevant to statistical downscaling, given that this technique works with sub-grid spatial scales (not resolved in our paper). In the interests of brevity, we think it is best to not speculate on this in the current manuscript.

Third, as mentioned in this paragraph, if the UK risk assessment is going to compare 4 and 2k, what are the implications for the climate projections needed to support such an assessment.

This is certainly a debate we will engage in, but the UK risk assessment was only intended as a brief example. There is not really space to give advice on how to apply the lessons from this paper to the various different applications that might be of interest to the broad readership of this journal.

The results are only consistent with the hypothesis (or suggestion) that non-linearities may be at play - you can't make a statement of evidence for non-linearities themselves unless you identify the actual non-linearity and also show that it is actually operating in one or more of the models.

In fact, we have shown in previous work that non-linearities exist in two Hadley Centre models, along with some initial look at mechanisms in HadGEM2-ES. We now cite these papers as a basis for this statement.

Whoops! Next time proof read a bit better before submitting.

Sorry! This error was introduced right at the end.

Methods

This methods section is only partially complete, with some of the other methods and approach described at the beginning of the results section. As this methods section so small, could it not be moved up as a stand alone section in the main text, and incorporating a general description of the analysis approach separate from the results section?

Thanks for this suggestion. We discussed this with the editor, who states that the current layout is as required by this journal.

Suggest you test the extent to which the results change if you reverse the historical periods.

Good point. We have tested this, and now state that, "Swapping the two historical periods in this analysis has no effect on the conclusions drawn from Figure 3 and Figure 4a (only minor changes in the pattern in Figure 4a are seen)."

Figures

Suggest you show the domains graphically on at least one of the maps in Figure 2.

Good point, thanks. We have done this on Figure 4c.

Reviewer #3 (Remarks to the Author):

Review of Good et al, Large differences in regional precipitation change between a first and second 2K of global warming, submitted to Nature Communications

This paper describes the projected rainfall changes for a 2K increase in global temperatures over pre-industrial via two scenario pathways, finding the pathway matters little to the precipitation response. It then contrasts these to a second 2K increase comparing the late 21st Century warming from RCP8.5 to that of the average of RCP2.6+RCP4.5, finding substantial non-linear responses in precipitation for this second 2K. It is a well-written and carefully crafted study using a novel approach with interesting results and I recommend publication.

Many thanks for the positive and helpful comments, which certainly improve the manuscript.

My only major comment is that while the paper provides much background into previous literature on non-linear rainfall responses, it has very little discussion of its results in direct comparison to recent studies assessing the robustness of pattern scaling techniques. In particular, a number of recent studies listed below have looked at the robustness of pattern scaling across policy relevant (RCP) scenarios and highlighted regions of model disagreement for precipitation. The results here need to be placed in better context with those studies.

Frieler, K., M. Meinshausen, M. Mengel, N. Braun, and W. Hare (2012), A scaling approach to probabilistic assessment of regional climate change, *J. Clim.*, 25(9), 3117-3144, doi:10.1175/JCLI-D-11-00199.1.

Tebaldi, C., and J. M. Arblaster (2014), Pattern scaling: Its strengths and limitations, and an update on the latest model simulations, *Clim. Change*, 122(3), 459-471, doi:10.1007/s10584-013-1032-9.

Herger, N., B. M. Sanderson, and R. Knutti (2015), Improved pattern scaling approaches for the use in climate impact studies. *Geophys. Res. Lett.*, 42, 3486-3494. doi: 10.1002/2015GL063569.

Yes, these references certainly need to be included. We now cite these in a number of places: in particular in an extended paragraph on pattern-scaling in the Discussion; but also an early mention in the introduction, stating that basic pattern scaling is known not to be exact. The Frieler paper is now also cited in the context of aerosol forcing and the China result.

Also, how does the presumed larger spread in global temperatures for the second 2K (since the larger values of warming to 4K will presumably lead to more model divergence around the ensemble mean) impact the results in Fig 3 and 4.

This is an interesting idea, but the model spread in global warming does not appear to be a factor in driving the reduced correlation in the right column of figure 3 and panel b of figure 4: the model standard deviation in global warming is quite similar for 4k-2k (mit) and 2k (fast) (0.31 and 0.37K). If the variability was different for these two intervals, for the reasons suggested by the reviewer, it

would still be classified as a nonlinear mechanism in our terminology, so for simplicity we prefer to not alter the text (detailed mechanisms will be addressed in future).

Minor comments:

Figure 1: 4K-2K(Mit) would be more understandable if relabelled to 4K(Fast)-2K(Mit)

We can see what the reviewer is getting at here, but there is a danger of adding other confusion. We prefer to not give 4K a distinguishing label because there is only one scenario that reaches 4K in the ensemble mean (rcp8.5).

Figure 3: the caption needs improving to note that the precipitation response is being plotted and units provided

Good point. Thanks.

Line 130-142: suggest moving these details after first introducing the experiments

Good point. We address this with a sentence giving a brief up-front introduction to the underlying experiments (and by moving the text as in the reviewer's next point). The first paragraph of the results now reads: "Here we focus on three different intervals in the CMIP5 simulations, that each show global, ensemble mean warming of close to 2K. These include fast and slow paths to 2K above pre-industrial conditions; and one interval from 2K to 4K above pre-industrial (in this case, the difference between a high-forcing and a mitigation scenario at the end of the century). For each of these 2K intervals, we examine the regional-scale differences in precipitation that accompany the 2K difference in global, ensemble mean warming."

Line 132: referring to Methods here is confusing as the experiments have yet to be introduced (and this is assumed knowledge in the Methods)

Agreed. We have moved these two sentences to the final paragraph of the section describing the 2K intervals.

Line 137: but pre-industrial conditions vary widely across models, particularly the magnitude of aerosol forcing used

The aim here was simply that, in each individual model, the changes in aerosol from that model's pre-industrial conditions to the end of the century should generally be smaller than if we examined changes from a recent historical period to the end of the century (in particular, aerosol forcings tend to return to close to pre-industrial conditions by the end of century in these rcps). Perhaps 'minimise' is an over-confident word here, so we have changed the text from 'We minimise pattern changes...' to 'We reduce pattern changes...' We also clarify at the start of the introduction that 'pre-industrial' refers to the CMIP5 pre-industrial control simulations.

Line 184: the main region of difference is China, not South East Asia (the latter being the countries to the south of China)

Good point - thanks. We have changed 'South East Asia' to China at each of the three times this is mentioned.

REVIEWERS' COMMENTS:

Reviewer #2 (Remarks to the Author):

The authors have done a good job of responding to all the reviews.
I suggest publication of the revised manuscript.
Thanks for an interesting read!

Reviewer #3 (Remarks to the Author):

I thank the authors for responding thoroughly to my comments and am pleased to recommend publication.